# Bond Behavior of Stainless-Steel and Ordinary Reinforcement Bars in Refractory Castables under Elevated Temperatures

Linas Plioplys [1], Andrius Kudžma [2], Aleksandr Sokolov [1], Valentin Antonovič [2] and Viktor Gribniak [1,*]

1   Laboratory of Innovative Building Structures, Vilnius Gediminas Technical University, Sauletekio av. 11,
    LT-10223 Vilnius, Lithuania; linas.plioplys@vilniustech.lt (L.P.); aleksandr.sokolov@vilniustech.lt (A.S.)
2   Laboratory of Composite Materials, Vilnius Gediminas Technical University, Linkmenu str. 28,
    LT-08217 Vilnius, Lithuania; andrius.kudzma@vilniustech.lt (A.K.); valentin.antonovic@vilniustech.lt (V.A.)
*   Correspondence: viktor.gribniak@vilniustech.lt; Tel.: +370-6-134-6759

**Abstract:** Refractory castables, i.e., refractory aggregates and ultra-fine particle mixtures with calcium aluminate cement (CAC) and deflocculants, were created 40 years ago for the metallurgy and petrochemical industries. These materials demonstrate outstanding performance even over 1000 °C. Typically, they have no structural reinforcement, resisting compression stresses because of the combination of temperature and mechanical loads. This study is a part of the research project that develops high-temperature resistance composite material suitable for fire and explosion protection of building structures. However, this application is impossible without structural reinforcement, and the bond performance problem becomes essential under high temperatures. This experimental work conducts pull-out tests of austenitic stainless 304 steel bars and typical structural S500 steel bars embedded in refractory castables after high-temperature treatments. This study includes plain and ribbed bars and considers two castable materials designed with 25 wt% CAC content for 50 MPa compressive strength after drying (typical design) and 100 MPa strength (modified with 2.5 wt% microsilica). This test program includes 115 samples for pull-out tests and 88 specimens for compression. As expected, the tests demonstrated the plain bars' inability to resist the bond stresses already at 400 °C; on the contrary, ribbed bars, even made of structural steel, could ensure a mechanical bond with cement matrix up to 1000 °C. However, only stainless steel bars formed a reliable bond with the high-performance castable, determining a promising object for high-temperature applications. Still, the scatter of the test results did not ensure a reliable bonding model. In addition, the castable strength might not be optimal to ensure maximum bond performance. Thus, the test results clarified the research objectives for further developing the reinforced composite.

**Keywords:** refractory castables; reinforcement bars; bond performance; stainless steel; pull-out tests; high temperature

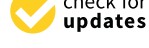



## 1. Introduction

Conventional refractory concrete with calcium aluminate cement (CAC), in which the content of calcium aluminates can range from 15 to 30%, is widely used in the thermal equipment linings of the energy industry [1,2]. This concrete stands out for its manufacturing simplicity: it can be produced on-site by mixing refractory aggregate with CAC. They also offer technological advantages, including easy blending, convenient shaping, and high mixture stability [2,3]. The conventional castables, incorporating CAC with aluminum oxide not exceeding 40% and aggregates with low $Al_2O_3$ content, stand out economically more attractive than medium- and low-CAC castables with expensive mixture components. The on-site manufacturing ability of conventional refractory materials further increases the economic benefits [3–5].

However, it is essential to consider a drawback of these concretes, which manifests as a deterioration of their mechanical properties after exposure to temperatures of 800 °C

to 1100 °C. This effect associates several essential interactions, including the binder's dehydration and the cement minerals' recrystallization at these elevated temperatures [6]. Due to these processes, the compressive strength of refractory concrete can decrease by about two times compared to its initial mechanical properties after production. Despite this limitation, conventional refractory concrete with CAC remains desirable in specific industrial applications where resistance to high temperatures is required and mechanical properties are not critical.

It has been established that modifying conventional refractory concrete with ultrafine $SiO_2$ micro-particles and deflocculants can enhance the properties of this concrete [7,8]. These $SiO_2$ micro-particles fill the gaps between concrete particles, participate in the hydration of CAC, and react with solid phases at high temperatures. Additionally, deflocculants reduce the water requirement in the concrete, thus reducing its porosity and imparting a denser structure [9].

Antonovich et al. [10] indicated that modified concrete containing these additives exhibits higher compressive strength, up to three times greater than conventional concrete without these additives. Wöhrmeyer et al. [7] also noted that not only do the mechanical properties of these modified concretes improve, but their thermal resistance and resistance to abrasion also increase. The cast-on binder is known for its excellent ability to maintain the material's mechanical strength at 600–1000 °C [11]. Furthermore, compared to traditional alternatives, the microscale silica/alumina activation process enhances castables' strength and sinterability temperature [12,13].

Despite the differences in their physical properties, refractory castables and ordinary Portland cement concrete share some similarities. Both materials are made of inorganic ceramic oxides used for structural purposes [14,15]. The reinforcement principles describe another similarity between these materials, although research on refractory materials focuses on corrosion problems [16].

Stainless steel can help prevent corrosion [17], and fiber reinforcement is a typical solution [18,19]. However, studies of refractory materials with bar reinforcements are rare. Andión et al. [20] provided a rare example of research on stainless-steel bars, although their study focused on corrosion problems. On the contrary, Bareiro et al. [18] conducted an extensive experimental program consisting of panel three-point bending and pull-out tests to determine the effects of fiber shape and elevated temperature on the mechanical performance of refractory castables. However, to the authors' knowledge, there has been no research on the bond performance of steel bars in refractory castables. This information is essential for developing reliable numerical models and efficient building structures where components of refractory material can protect the integrity of the building under fire and explosion actions [15,21,22].

This study belongs to a research project [23] developing fire protection systems with refractory protective shells. This manuscript investigates the bond performance of steel bars in refractory castables through pull-out tests of smooth and ribbed reinforcement bars. The test setup developed by Chu and Kwan [24] was adapted to investigate the effect of high temperature on the bond resistance of stainless steel and ordinary reinforcement bars. The test program used a conventional castable refractory with a target compressive strength and 25% weight of CAC. An alternative mixture is also modified with 2.5 wt% microsilica, which doubles the material's strength.

## 2. Materials and Methods

Two refractory castables containing 25 wt% of CAC ISTRA 40 with 40 wt% $Al_2O_3$ were selected for this study. The first composition is a conventional castable (CC0) with CAC and refractory aggregates. This mixture was proportioned to achieve a cold compressive strength of 50 MPa, i.e., the compressive strength, after drying the castable at 110 °C. The alternative composition (CC1) is the modified castable with 2.5 wt% microsilica, 2.5 wt% ground quartz sand, deflocculants, and the 100 MPa target cold compressive strength.

To prepare the refractory castables, the dry components were mixed for 5 min in a Hobart mixer, followed by adding water and another 5 min of mixing. The amount of water added was determined using the ASTM C0860-15R19 ball-in-hand test method [25]. Table 1 specifies the materials' proportions.

**Table 1.** Mix proportions of the refractory castable (wt%).

| Mix | CAC | Chamotte BOS145 | | SiO$_2$ | Milled Quartz Sand | Deflocculant CASTAMENT FS30 * | Water * |
| --- | --- | --- | --- | --- | --- | --- | --- |
| | | Crushed | Milled | | | | |
| CC0 | 25 | 60 | 15 | – | – | – | 14.3 |
| CC1 | 25 | 60 | 10 | 2.5 | 2.5 | 0.1 | 7.5 |

\* Over 100% dry mass.

This study focuses on the mechanical bond behavior of reinforcement bars in refractory castables. This investigation adopted the pull-out test setup developed by Chu and Kwan [24]. Pull-out tests were conducted on refractory samples prepared with 8 mm smooth and ribbed bars of the austenitic stainless 304 steel containing 18% chromium and 8% nickel. The alternative specimens had 8 mm ribbed bars from typical structural steel S500. Figure 1 shows the testing schematic and setup.

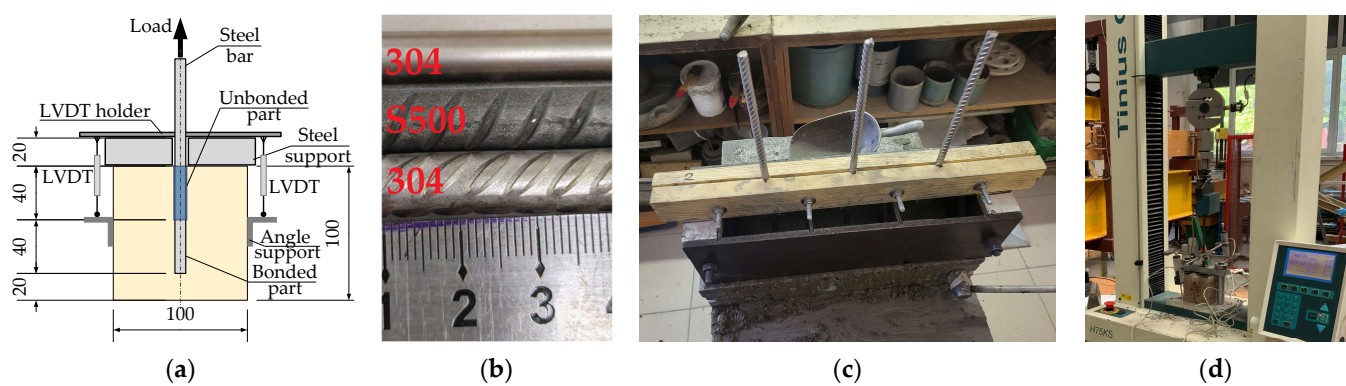

**Figure 1.** Pull-out test: (**a**) testing scheme; (**b**) bars; (**c**) the samples for pouring; (**d**) test setup.

The testing process involved a 110 °C reference and four treatment temperatures, i.e., 400 °C, 600 °C, 800 °C, and 1000 °C. The preparation and treatment (curing, drying, heat treatment) of the samples were carried out according to the requirements of the LST EN ISO 1927-5:2013 standard [26]. First, the castables were removed from the form after 72 h of curing at 20 ± 1 °C. After that, they were dried for 72 h at 110 ± 5 °C using a 2.0 kW drying camera. The samples were then heated for 5 h at the target temperatures using a 3.4 kW furnace with an electronic controller. The heating rate was 2.5 °C/min to 700 °C and 5.0 °C/min for temperatures ranging from 700 °C to 1000 °C. The reinforcement bar samples were heated with the castable specimens to determine the steel's post-heating (cold) mechanical performance. Five pieces of each bar type were subjected to each temperature.

The density and cold compressive strength (CCS) of the refractory castable 70 mm cube samples were determined following LST EN ISO 1927-6:2013 [27]. The density was estimated as the weight-to-volume ratio after subjecting the castable samples to drying at 110 °C and heat treatment at 400 °C, 600 °C, 800 °C, and 1000 °C; 46 CC0 samples and 42 CC1 samples were measured. The CCS tests were conducted on the same cube samples using the ALPHA 3-3000 S testing machine (FORM + TEST SEIDNER & CO., GmbH, Riedlingen, Germany).

The pull-out test samples included CC1 reinforced with smooth and ribbed stainless 304 steel and S500 ribbed bars and CC0 reinforced with 304 steel and S500 steel ribbed bars. Smooth-surface 304 steel bar samples for CC0 were not prepared because of insufficient

bond performance of the high-strength CC1 mixture with these bars. Table 2 specifies the number of specimens tested at each temperature. It demonstrates that the test campaign encompasses 115 cubes (Figure 1a) for pull-out tests and 88 specimens (70 mm cubes) for the density measurement and CCS tests. Figure 1b shows the bar surface shapes.

**Table 2.** The number * of the samples tested at different temperatures.

| Mix | Steel | Surface | 110 °C | 400 °C | 600 °C | 800 °C | 1000 °C |
|---|---|---|---|---|---|---|---|
| CC0 | 304 | Ribbed | 5/4 | 3/4 | 4/4 | 4/4 | 4/4 |
| | S500 | Ribbed | 4/5 | 5/5 | 4/5 | 4/5 | 3/6 |
| CC1 | 304 | Smooth | 5/– ** | 4/– ** | 4/– ** | 3/– ** | 4/– ** |
| | 304 | Ribbed | 4/4 | 3/3 | 4/3 | 4/4 | 4/4 |
| | S500 | Ribbed | 5(5)/4 ◇ | 8(3)/5 ◇ | 7(1 + 1)/5 ‡ | 10(4)/5 ◇ | 6(5)/5 ◇ |

\* The number describes "pulled-out samples/CCS samples". \*\* The same concrete mix as the S500 test. ◇ The number in the brackets indicates the bar failure without the pull-out consequence. ‡ The numbers in the brackets indicate the bar failure (one sample) and fracture of the concrete cube (one sample) without the pull-out consequence.

Figure 1c,d demonstrate the pull-out samples prepared for pouring and the test setup. The testing apparatus was a 75 kN capacity electromechanical machine H75KS (Tinius Olsen, Redhill, England) with a ±0.01% position measurement accuracy. The bar under testing was loaded in a deformation-control manner with a 2 mm/min loading rate. A 50 kN load cell measured the tension reaction with 0.5% precision. Two 50 mm linear variable displacement transducers (LVDTs) measured the relative displacement of the bar with 0.02% precision, as schematically depicted in Figure 1a. Thus, the following analysis considers the average value from two LVDT devices. Readings from all devices (LVDTs and the load cell) were acquired every second through the signal processing equipment Almemo 2890-9 and recorded by a workstation computer.

## 3. Results and Discussion

### 3.1. Materials Properties

Figure 2a summarizes the density analysis results and indicates the scatter of the results expressed in the standard deviation terms. Table 2 (the number under the slash) describes the test sample quantity. Figure 2a shows the 2327 kg/m$^3$ density of CC1 after drying at 110 °C. Compared to the CC0 result (2124 kg/m$^3$), the material's structure densification reached 9.6%. A similar trend persists even after the thermal treatments. These results align with observations by Lee et al. [8], who found that incorporating $SiO_2$ micro-particles and a deflocculant into conventional concrete increases the density of refractory concrete. This phenomenon is attributed to $SiO_2$ micro-particles, which, in addition to filling the gaps between particles, participate in the hydration of CAC, thereby contributing to a denser material structure. The $SiO_2$ micro-particles reduce material porosity [9], and the reduced water content further enhances this effect. Hydrothermal conditions also affect the hydration process and the remaining clinker phase reaction [28,29].

Figure 2b shows the CCS test results and, in the same manner as Figure 2a, indicates the scatter of the results corresponding to the sample quantity listed in Table 2; the CCS tests used the same samples as the density tests. Figure 2b demonstrates that the target strength was achieved in both castables. In conventional concrete CC0, the strength after drying at a temperature of 110 °C reached 45.9 MPa but dropped nearly by half (25.6 MPa) after treatment at 1000 °C. This phenomenon can be linked to the literature, which suggests that conventional concrete dehydrates up to 400 °C [30] and starts forming the $C_{12}A_7$ cement mineral from amorphous dehydrated calcium aluminate to 900 °C [31]. Figure 2b illustrates this trend—the CC0 strength drops to 35.8 MPa after 400 °C. At 600 °C and 800 °C, the tests demonstrate no significant decrease in strength. However, from 900 °C to 1000 °C, $C_{12}A_7$ cement minerals begin to react with aluminum and form elongated CA minerals [8,31]; thus, the strength of conventional concrete drops further to 25.6 MPa.

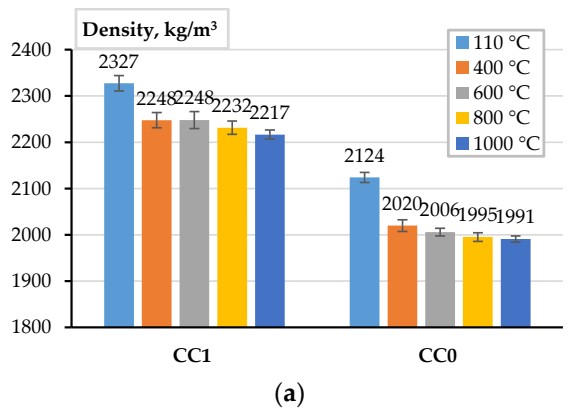

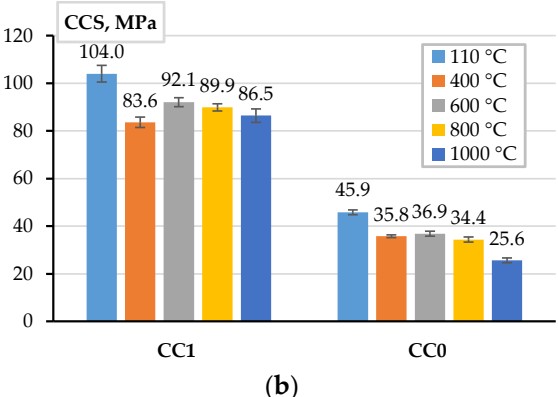

**(a)**       **(b)**

**Figure 2.** Material characteristics after different temperature treatments: (**a**) density; (**b**) cold compressive strength (CCS).

A similar trend is observed with the modified CC1 concrete. The specimens, dried at 110 °C, reached a 104.0 MPa strength, which decreased to 86.5 MPa after treatment at 1000 °C. Figure 2b shows a 17% drop in the CC1 strength, underperforming the 50% decrease in the compression resistance of the CC0 samples. Overall, the increased compression performance of the modified CC1 castable regarding CC0 specimens results from adding $SiO_2$ micro-particles. These particles reduce specimen porosity [9] and react with cement hydrates, forming new hydrates such as stratlingite ($C_2ASH_8$) [11]. The stratlingite undergoes dehydration and becomes amorphous at 210 °C [12]. Thus, it does not participate in recrystallization at higher temperatures (1000 °C), restricting the formation of $C_{12}A_7$ and CA minerals and reducing the decrease in strength.

Figure 3 shows the results of the tensile test of reinforcement bars after the high-temperature treatments; the 'Reference' diagrams correspond to the unheated samples. This figure shows an insignificant decrease in the mechanical performance of all steel samples until the treatment conditions reach 600 °C. This outcome aligns with the literature results [32–35]. The further temperature increase caused a substantial degradation in the strength of S500 steel samples: a 50% and 75% reduction in yield strength corresponds to 800 °C and 1000 °C treatment; Plioplys et al. [36] reported the analysis details. At the same time, a more ductile response is characteristic of stainless steel 304. The ultimate deformations of the 304 steel also increase with the temperature, exceeding 0.4 strain after 1000 °C heating. For comparison, the maximum strain only slightly exceeds 0.06 for the S500 steel. The maximum residual strength of the stainless steel exceeds 600 MPa, making it an excellent alternative for reinforcing structures subjected to elevated temperatures.

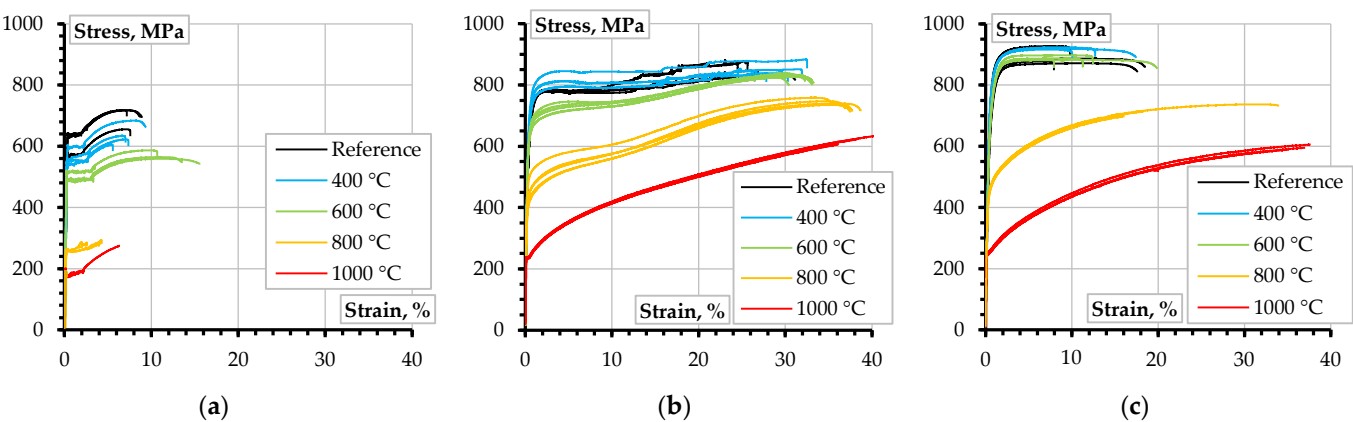

**(a)**       **(b)**       **(c)**

**Figure 3.** Tensile test results of 8 mm bar reinforcement: (**a**) ribbed S500 steel bars; (**b**) smooth stainless-steel bars; (**c**) ribbed stainless-steel bars.

*3.2. Pull-Out Tests*

Figure 4 shows the pull-out test results, indicating the following essential aspects:

- *Ribbed S500 steel bars in CC1 cubes.* The bond of the reference CC1 samples (heated at 110 °C) was too strong to allow for pulling-out failure of the specimens—the reinforcement breakage was the test consequence. Therefore, Figure 4a does not include the 'Reference' diagrams. In addition, reinforcement failure was observed at all temperature ranges, making S500 steel unsuitable for structural use in combination with the modified CC1 castable.

- *Ribbed S500 steel bars in CC0 cubes.* Figure 4b shows the expected results for the specimens until 400 °C—the bond resistance is proportional to the compression test results in Figure 2b. With the temperature reaching 800 °C, the bond strength does not change. At the same time, the CC0 samples' failure remains ductile regarding CC1 specimens (Figure 4a). Figure 4b shows that the bond performance of the CC0 cubes drastically decreases after exceeding the 10 mm pull-out deformation.

- *Smooth stainless 304 steel bars in CC1 cubes.* Figure 4c shows a substantial bond strength reduction after heating the samples already at 400 °C. After a 1000 °C treatment, the concrete lost contact with reinforcement, and the bars were pulled out by hand. Therefore, Figure 4c does not include diagrams corresponding to the temperature of 1000 °C. In other words, the composite effect in the specimens reinforced with plain bars disappeared after heating, making it inefficient for structural application.

- *Ribbed stainless 304 steel bars in CC1 cubes.* Figure 4e shows that combining the ribbed bars with the modified mixture of CC1 resulted in the best bonding performance, almost doubling the pull-out resistance compared to the smooth bars (Figure 4c) and CC0 cubes (Figure 4d). Although the stainless 304 steel results are comparable to S500 in the CC1 cubes (Figure 4a), the higher tensile strength of the stainless 304 steel (Figure 3c) compared to the structural S500 bar (Figure 3a) prevents the bar failure. The deformation increase in Figure 4a regarding Figure 4d results from the yielding of S500 steel. Thus, the ductile debonding of the stainless 304 steel shows the practical application possibility of such composites at elevated temperatures.

- *Ribbed stainless 304 steel bars in CC0 cubes.* Figure 4e shows that the ribbed bars significantly improve the bonding performance compared to the smooth bars (Figure 4c). However, the ultimate resistance of the S500 bars in the same concrete (Figure 4b) was not reached. That is a consequence of the different shapes of the reinforcement ribs (Figure 1b). However, the improved mechanical performance of the modified castable CC1 also improves the bond performance (Figure 4d), making the latter combination promising for structural use.

The shear stresses acting through the bond are essential for numerical modeling and analysis of the composite behavior [37–42]. Therefore, this study approximates the characteristic points schematically depicted in Figure 5. The following expression describes the average shear stresses:

$$\tau = \frac{P}{\pi \cdot \varnothing \cdot l},\tag{1}$$

where $P$ is the axial load; $\varnothing$ is the bar diameter (assumed equal to 8 mm for all bars); $l$ is the bonding length (40 mm in the considered case).

Table 3 gives the approximation results for the ribbed bars for both castables, indicating the average characteristic values and corresponding standard deviations. This analysis includes only pulled-out cases, i.e., the first numbers in Table 2, subtracting the figures in the brackets. The results of Table 3 support the conclusions from Figure 4. The essential aspect is that ordinary bars can resist bond stresses even after thermal treatment, outperforming the stainless-steel counterparts in CC0. That is the apparent consequence of the shape of the ribs optimized for structural use (Figure 1b). These observations align with the results of the literature [43,44]. However, the strength increase (regarding the reference concrete CC0, Figure 2b) made the ribbed stainless-steel bars efficient in the modified castable

CC1. The smooth ribs also control the bond performance and increase the ductility of the reinforcement system. Still, literature [45] considered stainless-steel bars with a sharp rib geometry, which caused splitting concrete failure during the pull-out tests. However, this outcome could result from the insufficient strength of the concrete.

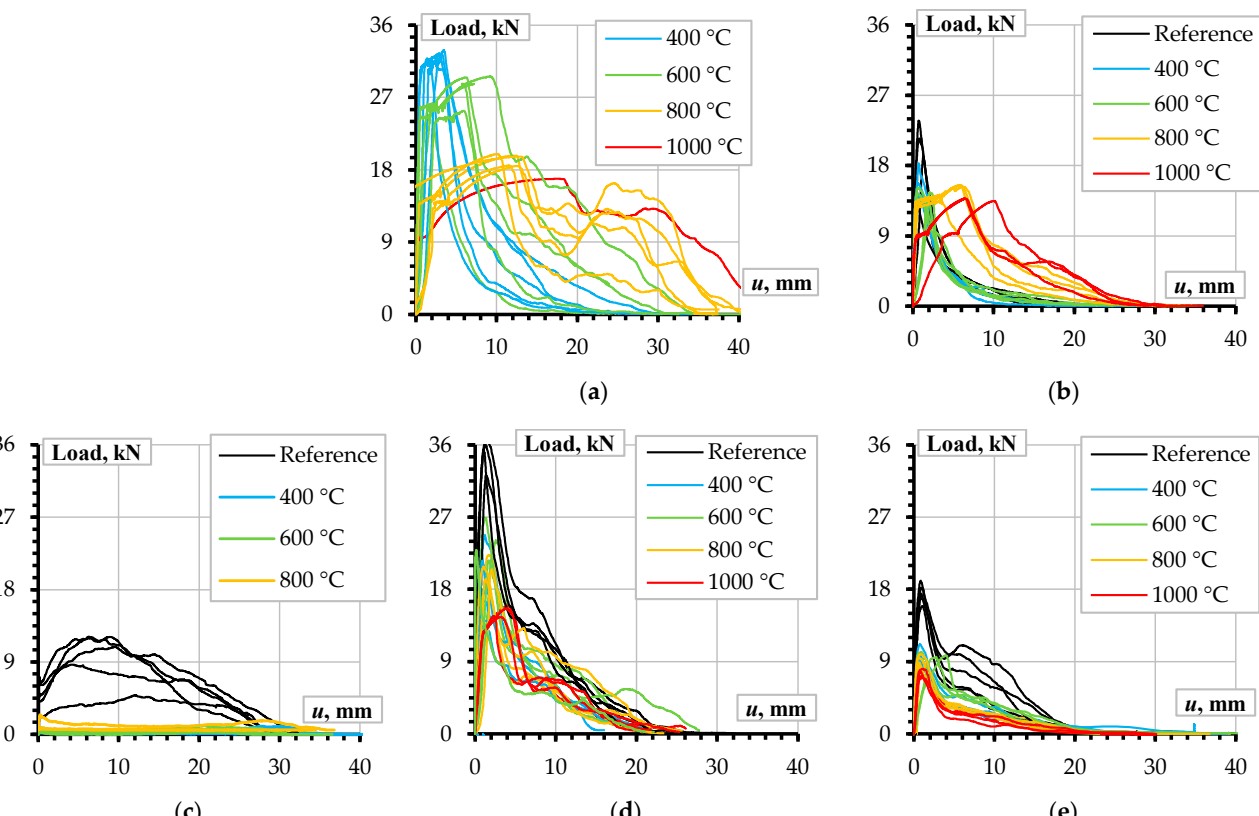

**Figure 4.** Pull-out results: (**a**) S500 steel ribbed bar from CC1; (**b**) S500 steel ribbed bar from CC0; (**c**) 304 steel smooth bar from CC1; (**d**) 304 steel ribbed bar from CC1; (**e**) 304 steel ribbed bar from CC0.

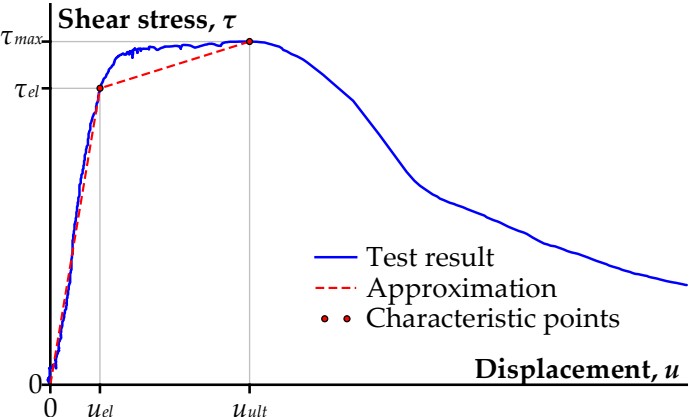

**Figure 5.** Characteristic points of the shear stress–displacement diagram.

At the same time, Table 3 shows a substantial scatter of the test results expressed in the standard deviation terms. The most significant variation is characteristic of the elastic stage boundaries (Figure 5). The scatter of elastic stress ($\tau_{el}$) and deformation ($u_{el}$) values sometimes exceeded 50%. The ultimate stress ($\tau_{max}$) variation is less significant and typically does not exceed 10%. Still, the corresponding deformation ($u_{ult}$) variation well exceeded 50%.

**Table 3.** Characteristic values in Figure 5 under different temperatures (mean ± standard deviation).

| Steel | Mix | Parameter | 110 °C | 400 °C | 600 °C | 800 °C | 1000 °C |
|---|---|---|---|---|---|---|---|
| S500 | CC0 | $\tau_{el}$ (MPa) | 13.23 ± 6.54 | 13.75 ± 1.80 | 9.20 ± 3.01 | 11.90 ± 1.07 | 6.87 ± 0.25 |
| | | $\tau_{max}$ (MPa) | 20.63 ± 2.04 | 15.57 ± 1.66 | 14.55 ± 0.36 | 14.97 ± 0.64 | 11.18 ± 3.39 |
| | | $u_{el}$ (mm) | 0.73 ± 0.22 | 1.08 ± 0.09 | 0.85 ± 0.45 | 0.53 ± 0.19 | 1.38 ± 1.71 |
| | | $u_{ult}$ (mm) | 0.86 ± 0.26 | 0.62 ± 0.08 | 0.75 ± 0.40 | 5.26 ± 1.28 | 6.35 ± 0.14 |
| | CC1 | $\tau_{el}$ (MPa) | – | 12.70 ± 5.01 | 12.62 ± 7.76 | 10.04 ± 1.5 | 9.25 * |
| | | $\tau_{max}$ (MPa) | – | 31.29 ± 1.67 | 26.92 ± 1.76 | 19.16 ± 0.58 | 16.82 * |
| | | $u_{el}$ (mm) | – | 0.50 ± 0.34 | 0.44 ± 0.31 | 1.74 ± 1.58 | 0.3 * |
| | | $u_{ult}$ (mm) | – | 2.36 ± 1.12 | 5.77 ± 2.33 | 11.06 ± 2.09 | 17.36 * |
| 304 | CC0 | $\tau_{el}$ (MPa) | 12.92 ± 3.50 | 8.60 ± 0.43 | 7.18 ± 0.88 | 5.37 ± 0.69 | 6.15 ± 0.81 |
| | | $\tau_{max}$ (MPa) | 17.42 ± 1.06 | 10.18 ± 0.75 | 9.59 ± 0.17 | 9.34 ± 0.42 | 7.50 ± 0.40 |
| | | $u_{el}$ (mm) | 0.48 ± 0.20 | 0.43 ± 0.05 | 0.95 ± 0.50 | 0.30 ± 0.06 | 0.63 ± 0.15 |
| | | $u_{ult}$ (mm) | 0.91 ± 0.08 | 1.20 ± 0.92 | 0.96 ± 0.08 | 0.67 ± 0.07 | 0.95 ± 0.11 |
| | CC1 | $\tau_{el}$ (MPa) | 27.6 ± 4.21 | 19.58 ± 0.70 | 20.83 ± 0.89 | 17.85 ± 1.76 | 10.33 ± 1.18 |
| | | $\tau_{max}$ (MPa) | 34.62 ± 1.89 | 21.66 ± 2.48 | 22.37 ± 4.01 | 19.91 ± 0.67 | 15.13 ± 0.62 |
| | | $u_{el}$ (mm) | 0.80 ± 0.37 | 0.59 ± 0.14 | 1.03 ± 0.64 | 1.10 ± 0.41 | 0.75 ± 0.11 |
| | | $u_{ult}$ (mm) | 1.31 ± 0.20 | 0.78 ± 0.33 | 1.42 ± 0.87 | 1.42 ± 0.26 | 3.34 ± 0.70 |

* A single sample was successfully pulled out during the tests.

The results of Figure 4 and Table 3 indicate the low reliability of the bond performance assessments, proclaiming the need for developing more reliable bonding solutions. In this context, combining the austenitic stainless 304 steel bars with a ribbed surface and modified CC1 castable describes a promising solution for optimizing the bond properties of reinforced composites subjected to elevated temperatures.

### 3.3. Bond Failure Mechanisms

The pull-out outcomes of the conventional castable CC0 are not surprising and result from the material performance reduction mechanisms discussed in Section 3.1. Thus, the rib shapes controlled the bond performance while the concrete strength degraded under temperature impact. However, the castable modification (CC1) alters the bonding mechanisms. As Table 2 shows (numbers in the brackets), 18 pull-out CC1 samples reinforced with the S500 bar faced a brittle collapse because of the yielding of the steel bar (Figure 6a) and one cube crushing (Figure 6b). Figure 6c shows the "successful" pull-out test consequence. The probability distribution of brittle failure was somewhat evenly spread across different temperatures, varying from 0.3 to 0.4 (Table 2). Still, all reference samples (after 110 °C) experienced brittle failure, and only one bar was pulled out after 1000 °C treatment. The failure of the reference samples resulted from excessive concrete strength for activating the bond-slip mechanisms prevalent for structural applications [32]. The 1000 °C treatment reduces the mechanical performance of S500 steel (Figure 3a), making it inapplicable for structural use.

Figure 7 compares the pulled-out bar surfaces after 1000 °C. Figure 7a,c show the ribs of S500 and 304 steel bars under 50× magnification. The differences are not apparent. However, the 500× magnification (of the same rib) clarifies the structure changes. In particular, Figure 7b shows the iron carbonade crystals forming a brittle layered structure at the bar surface [46,47].

Figure 6a exemplifies the debonding of the carbonization layers formed at the S500 bar surface after 800 °C treatment, causing the bar failure. Cao et al. [47] mentioned the performance of the concrete to passivate the iron carbonization process. However, Figure 7b shows the apparent carbonization signs inside the pull-out sample—this rib was inside the concrete during the heating process. On the contrary, Figure 7d shows the stable surface structure when the concrete remained attached to the bar surface. This image could explain the outstanding bonding performance of stainless-steel bars in the modified castable CC1. It also allows the hypothesis that the microsilica stimulated the hydration process at the bar surface. However, additional tests are necessary to verify this expectation.

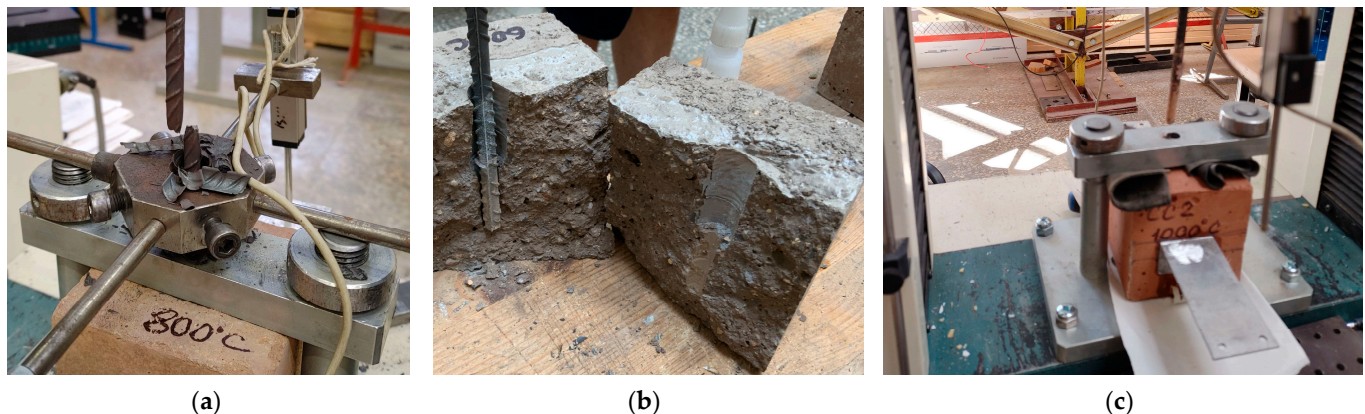

**Figure 6.** Pull-out test results of CC1 with S500 reinforcement: (**a**) yielding of S500 bars (18 samples); (**b**) failure of the concrete (one cube); (**c**) the pulled-out bar (97 elements).

(**a**)

(**b**)

(**c**)

(**d**)

**Figure 7.** SEM images of the bar surface (rib) treated at 1000 °C after the pull-out test: (**a**,**b**) S500 steel under 50× and 500× magnification; (**c**,**d**) 304 steel under 50× and 500× magnification.

*3.4. Further Investigation*

The results of this study indicate that the temperature treatment substantially affects the bond performance of steel bars in refractory castables, and pull-out tests are a suitable method to estimate this effect. The experiments demonstrated that ribbed bars are more efficient than their smooth counterparts in resisting mechanical loads after elevated temperatures because of the mechanical rib interlock with the surrounding concrete. However, a brittle fracture occurred when S500 steel reinforcement was inserted into the modified high-strength CC1 castable, making this reinforcement system unsuitable for structural purposes [48]. Furthermore, despite some promising results at elevated temperatures, the S500 reinforcement strength was insufficient to activate the bond-slip mechanism before heating (Figure 4a). In addition, a brittle splitting of the CC1 cube was observed after 600 °C (Figure 6b), which could result from the different thermal expansion of steel and concrete and stress concentration in the castable because of the sharp shape of the ribs of the S500 bars (Figure 1b) that are efficient in ordinary concrete but irrelevant for the elevated temperature conditions [49]. Still, a sharp rib geometry is also characteristic of stainless-steel bars [45] that could raise stress concentrations, deteriorating the bond with concrete.

On the contrary, the cold-formed ribs of the stainless 304 steel bars in this study were efficient with the modified CC1 castable, ensuring the ductile pull-out mechanism (Figure 6c), which is acceptable for structural composites. The SEM images (Figure 7) put forward the bond defragmentation mechanisms of S500 bars, providing promising insights for the austenitic stainless 304 steel reinforcement.

As part of the research project [23], this study highlighted the potential of stainless 304 steel ribbed bars for developing protective refractory shells. Still, reducing the scatter of the pull-out test results (Figure 4d and Table 3) is needed for a reliable bond model. In addition, the concrete strength might not be optimal to ensure maximum bond performance; the mechanisms behind the contact surface improvement (Figure 7d) must be clarified. So, these aspects determine the objectives for further research. In addition, the structural performance of reinforced systems, e.g., as described in reference [48], under elevated temperatures needs evaluation.

## 4. Conclusions

This experimental study investigates the bonding performance of smooth and ribbed reinforcement bars in refractory concretes. The experimental campaign encompasses two calcium aluminate cement (CAC) castables, i.e., the conventional mixture with a 50 MPa compression strength and the modified castable with 100 MPa strength; the 8 mm reinforcement bars from austenitic stainless 304 steel (smooth and ribbed surface) and structural S500 steel (ribbed surface) were used for the pull-out tests. The tests characterized the bond resistance after temperature treatment at 110 °C, 400 °C, 600 °C, 800 °C, and 1000 °C. The program included 115 cubes (100 mm in size) for pull-out tests and 88 samples of 70 mm cubes for the cold compression tests. The following conclusions are made:

- The mechanical interlock ensures reinforcement to the refractory material after high-temperature heating. Structural S500 steel bars demonstrated bonding ability with the modified castable even after heating at 1000 °C. However, these samples' brittle fracture was characteristic, demonstrating this combination's irrelevance for structural applications. Moreover, the SEM analysis determined the iron carbonization signs at the bar surface inside the concrete, which also reduces the bond strength.
- Replacing the S500 ribbed bars with stainless 304 steel ribbed bars was efficient. In particular, the stainless-steel ribbed bars showed good pull-out results comparable to S500 and CC1 under elevated temperatures but without brittle failure risks. However, high temperatures still significantly affect bond performance, and there is room for improvement in the refractory concrete mixture.
- The smooth surface stainless-steel bars demonstrated a substantial reduction in bond strength after 400 °C and complete loss of contact with concrete after 1000 °C. This re-

sult proclaims the inability of these bars to reinforce concrete elements under elevated temperatures.

- The scatter of the bond properties sometimes exceeding 50% makes the mechanical characteristics unreliable, proclaiming the necessity of developing a more reliable bonding system. This study revealed a promising potential of combining the stainless-steel ribbed bars with modified CC1 castable, the optimization of which describes a further research object.

**Author Contributions:** Conceptualization, methodology, and validation, V.G. and V.A.; software, formal analysis, L.P.; investigation, L.P., A.S. and A.K.; resources, V.G.; data curation and writing—original draft preparation, L.P. and A.K.; writing—review and editing, V.G.; visualization, L.P. and A.K.; project administration, A.S.; supervision and funding acquisition, V.G. All authors have read and agreed to the published version of the manuscript.

**Funding:** This project has received funding from the European Regional Development Fund (Project No. 01.2.2-LMT-K-718-03-0010) under a grant agreement with the Research Council of Lithuania (LMTLT).

**Data Availability Statement:** The authors will provide the raw data of this work upon request.

**Conflicts of Interest:** The authors declare no conflict of interest.

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
