# Peer review of "Bond Behavior of Stainless-Steel and Ordinary Reinforcement Bars in Refractory Castables under Elevated Temperatures"

_jcs, doi:10.3390/jcs7120485_

Round 1

Reviewer 1 Report

Comments and Suggestions for Authors

This contribution belongs to a research project developing fire protection systems with refractory protective shells. The present study aimed to investigates the bond performance of steel  bars in refractory castables through pull-out tests of smooth and ribbed reinforcement bars. The test setup was adapted to investigate the effect of high temperature on the bond resistance of stainless steel and ordinary reinforcement bars. The test program used a conventional castable refractory with a target compressive strength and 25% weight of CAC. An alternative mixture is also modified with 2.5wt% microsilica, which doubles the material strength. Generally, it explains what has been done and what has been discovered. The subject matter is interesting because the paper gives access to the information of this particular research. In summary, this is a piece of acceptable work but it needs some revisions. The paper can be accepted for publication, provided that the following points can be clarified.

1. The abstract should reflect the purpose of the study, the main results, the advantages of the approach used. In addition, nothing is said about the experiment, although much attention is paid to the description of the experiment in the manuscript.

2. Details of  test rig components -  technical features, measuring ranges and accuracy of the measuring devices of these instruments in Table should be given.

3. The conclusions can be further elaborated:  what was the limitation of the study?, what do the study results imply for the larger stakeholders? (policy makers, practitioners, the economic sector, etc. address them one by one),  what future study avenues are opened / suggested after this study?

4. Please provide the statistical test results. Please indicate how many rounds of trials were conducted? Please also report their variances. Please validate the proposed method on multiple datasets. 

5. The decision tree has the following assessment criteria, prediction accuracy, over-learning, complexity, processing scale and robustness. The authors should discuss based on these indicators.

Author Response

This contribution belongs to a research project developing fire protection systems with refractory protective shells. The present study aimed to investigates the bond performance of steel bars in refractory castables through pull-out tests of smooth and ribbed reinforcement bars. The test setup was adapted to investigate the effect of high temperature on the bond resistance of stainless steel and ordinary reinforcement bars. The test program used a conventional castable refractory with a target compressive strength and 25% weight of CAC. An alternative mixture is also modified with 2.5wt% microsilica, which doubles the material strength. Generally, it explains what has been done and what has been discovered. The subject matter is interesting because the paper gives access to the information of this particular research. In summary, this is acceptable work, but it needs some revisions. The paper can be accepted for publication, provided the following points can be clarified.

Answer: The authors wish to express their sincere gratitude to the Reviewer for the positive evaluation of the submission. The manuscript has been improved by implementing all suggestions by the Reviewer – the modifications in the text are highlighted in yellow.

  1. The abstract should reflect the purpose of the study, the main results, and the advantages of the approach used. In addition, nothing is said about the experiment, although much attention is paid to the description of the experiment in the manuscript.

Reply: The Authors accepted the criticism and clarified the essential contribution of this experimental study. Still, they did not shorten the introduction part of the Abstract, fearing losing the presentation consistency.

Correction in the manuscript: The newly introduced last two sentences of the Abstract clarify the essential contribution of this experimental program.

  1. Details of test rig components - technical features, measuring ranges, and accuracy of the measuring devices of these instruments in Table should be given.

Reply: The Authors sincerely appreciate this note, highlighting the essential information missed in the original manuscript. They updated the presentation correspondingly.

Correction in the manuscript: The new paragraph (Lines 131–140) describes the measurement procedure and precision.

  1. The conclusions can be further elaborated: what was the limitation of the study? What do the study results imply for the larger stakeholders? (policymakers, practitioners, the economic sector, etc. address them one by one), what future study avenues are opened/suggested after this study?

Reply: The Authors are grateful for this thoughtful comment. Remarkably, the obtained results are far away from the direct interests of policymakers. As part of the ongoing research project, which creates fire-protection structural systems, this study highlighted the potential of the austenitic stainless 304 steel ribbed bars for developing protective refractory shells. Still, reducing the scatter of the pull-out test results is needed for a reliable bond model. In addition, the concrete strength might not be optimal to ensure maximum bond performance; the mechanisms behind the contact surface improvement must be clarified. So, these aspects determine the objectives for further research. The Authors added the corresponding comment in Section 3.4 of the updated manuscript.

Correction in the manuscript:

  • The title of Section 3.4 was modified to “Further investigation.”
  • The new paragraph (Lines 327–334) was introduced to clarify this study’s contribution.

  1. Please provide the statistical test results. Please indicate how many rounds of trials were conducted? Please also report their variances. Please validate the proposed method on multiple datasets.

Reply: First, this study considers a new application of refractory castables, which typically have no structural reinforcement (e.g., Line 13 of the updated manuscript). Therefore, understanding the bonding mechanisms becomes paramount for developing structural fire-protection systems. This situation results in the absence of suitable alternative datasets. The Authors believe that the updated Abstract and Introduction clarify this problem.

This preliminary study (see reply to the above comment) includes 115 samples for pull-out tests and 88 specimens for compression. Table 2 describes the corresponding quantities of the test samples. All resultant diagrams and tables demonstrate the scatter of results of the nominally identical samples. Where possible (meaningful), the standard deviations accompany the average values (e.g., Figure 2 and Table 3); where not, the alternative diagrams are presented (e.g., Figures 3 and 4). The latter outcome resulted from substantial scatter of the test results. This output aligns with the literature results, e.g., [https://doi.org/10.1007/s11029-014-9432-0; https://doi.org/10.1061/(ASCE)ST.1943-541X.0002217]. Therefore, the Authors see no necessity to complicate the presentation with irrelevant discussions. Still, they clarified the contribution of this study, replying to the above comment. At the same time, the Authors understood that the relationship between Tables 2 and 3 is not apparent. Therefore, they added the corresponding comments in the text.

Correction in the manuscript:

  • Figure 2. The following comments were added in Lines 154–156 and 165–167: “Figure 2a summarizes the density analysis results and indicates the scatter of the results expressed in the standard deviation terms. Table 2 (the number under the slash) describes the test sample quantity. <…> Figure 2b shows the CCS test results and, in the same manner as Figure 2a, indicates the scatter of the results corresponding to the sample quantity listed in Table 2; the CCS tests used the same samples as the density tests.”
  • Table 3. The following clarification was added in Lines 246–249: “Table 3 gives the approximation results for the ribbed bars for both castables, indicating the average characteristic values and corresponding standard deviations. This analysis includes only pulled-out cases, i.e., the first numbers in Table 2, subtracting the figures in the brackets.”

  1. The decision tree has the following assessment criteria, prediction accuracy, over-learning, complexity, processing scale and robustness. The authors should discuss based on these indicators.

Reply: The Authors appreciate this note. However, they believe that the improvement made in replying to Comment 3 above clarifies the research idea and essential contribution.

Reviewer 2 Report

Comments and Suggestions for Authors

1.       Figure 2. Test setup. The icon introduction is too simple, the specific performance test content is not introduced, and no relevant description is found in "materials and methods". I recommend merging Figure 1 and 2 into one picture.

2.       "Figure 3a supports this observation." It is recommended that when describing results, you first describe the content of the picture, and then explain the content or even discuss the content. Then quote other people's opinions to prove or explain your own phenomenon.

3.       It is recommended to crop out the shooting information below and add a ruler in Figure 8.

4.       "3.4. Concluding note" should be written after each result. Or write results and discussion separately.

Author Response

  1. Figure 2. Test setup. The icon introduction is too simple, the specific performance test content is not introduced, and no relevant description is found in "materials and methods". I recommend merging Figures 1 and 2 into one picture.

Reply: The Authors sincerely appreciate the noted drawback and merged these figures as suggested.

Correction in the manuscript:

  • Figures 1 and 2 were merged.
  • The Authors added the reference to Figure 1d (former Figure 2) in the text and discussed the testing procedure.

  1. "Figure 3a supports this observation." It is recommended that when describing results, you first describe the content of the picture and then explain the content or even discuss the content. Then quote other people's opinions to prove or explain your own phenomenon.

Reply: The Authors acknowledge this thoughtful note and agree that the original description was unclear.

Correction in the manuscript: The structure of the noted paragraph was changed as recommended.

  1. It is recommended to crop out the shooting information below and add a ruler in Figure 8.

Reply: The Authors accepted the recommended modification. Still, they did not crop the bottom information typical for SEM images.

Correction in the manuscript: The scales were added to the SEM images.

  1. "3.4. Concluding note" should be written after each result. Or write results and discussion separately.

Reply: The comment is sincerely appreciated. The Authors intended to summarize the findings and formulate further research prospects. However, they agree that the original section was misleading. Furthermore, this criticism aligns with the recommendation of another Reviewer. Therefore, the Authors rephrased the last paragraph and added one paragraph to clarify this study’s contribution to the current knowledge state.

Correction in the manuscript:

  • The title of Section 3.4 was modified to “Further investigation.”
  • The text (the last paragraph of Section 3.4) was modified to clarify this study’s contribution.

Acknowledgment. The authors sincerely thank the Reviewer for sharing time and knowledge. Comments and suggestions that have contributed to the improvement of the manuscript are genuinely appreciated. The yellow color highlights all corrections in the text.

Round 2

Reviewer 1 Report

Comments and Suggestions for Authors

I agree with the responses of the authors and recommend publishing the contribution in the submitted wording

Reviewer 2 Report

Comments and Suggestions for Authors

The manuscript has been revised properly. It can be accepted.